# Framework with cytoskeletal actin filaments forming insect footpad hairs inspires biomimetic adhesive device design

Ken-ichi Kimura [1]✉, Ryunosuke Minami[1], Yumi Yamahama[2], Takahiko Hariyama [3] & Naoe Hosoda [4]✉

Footpads allow insects to walk on smooth surfaces. Specifically, liquid secretions on the footpad mediate adhesiveness through Van der Waals, Coulomb, and attractive capillary forces. Although the morphology and function of the footpad are well defined, the mechanism underlying their formation remains elusive. Here, we demonstrate that footpad hair in *Drosophila* is formed by the elongation of the hair cells and assembly of actin filaments. Knockdown of *Actin5C* caused a malformation of the hair structure, resulting in reduced ability to adhere to smooth substrates. We determined that functional footpads are created when hair cells form effective frameworks with actin filament bundles, thereby shaping the hair tip and facilitating cuticular deposition. We adapted this mechanism of microstructure formation to design a new artificial adhesive device—a spatula-like fiber-framed adhesive device supported by nylon fibers with a gel material at the tip. This simple self-assembly mechanism facilitates the energy-efficient production of low-cost adhesion devices.

[1] Laboratory of Biology, Sapporo Campus, Hokkaido University of Education, Sapporo 002-8502, Japan. [2] Department of Biology, Hamamatsu University School of Medicine, Hamamatsu 431-3192, Japan. [3] Institute for NanoSuit Research, Preeminent Medical Photonics Education and Research Center, Hamamatsu University School of Medicine, Hamamatsu 431-3192, Japan. [4] Research Center for Structural Materials, National Institute for Materials Science, Tsukuba 305-0044, Japan. ✉email: kimura.kenichi@s.hokkyodai.ac.jp; HOSODA.Naoe@nims.go.jp

nsects that can walk on smooth surfaces have specialized structures called footpads on their legs[1,2]. Previous studies have shown that footpads are composed of flexible hairs (setae), the tips of which have specific microstructures that are spatulate, discoidal, or pointed[3–5]. The footpad generates Van der Waals, Coulomb, and attractive capillary forces facilitating adhesion on to the substrate surface[6]. The footpad secretion is known to contain a non-volatile, lipid-like substance[2,7,8]. The bridge between the setal tip and the substrate, filled with the secreted liquid generates capillary forces that allow the insects to adhere to the substrate[6]. Although the morphology and function of the footpad are relatively understood[2], the mechanism underlying its formation remains elusive. *Drosophila melanogaster* is a very useful model insect to study the mechanism of the development of footpads owing to the availability of several advanced genetic tools. The *Drosophila melanogaster* footpad houses many hairs, the tips of which have a spatula-like microstructure[9,10]. Recently, a mutant of the Polycomb group gene *Su(z)2* with malformed adhesive pad structures was reported to affect the insect's climbing ability[10]. However, the molecular and cellular mechanisms underlying footpad formation remain largely unknown.

In holometabolous insects such as *Drosophila*, footpads are newly formed during metamorphosis[11,12]. Here, we examine the formation of the footpad during the pupal stage. During the formation of the footpad hair, a hair cell elongated and actin filaments are assembled to make a framework providing a specific shape for the hair. Inhibition of cytoskeletal actin function through the RNAi-mediated knockdown of a gene *Actin5C (Act5C)* induced the malformation of the hair structure. The *Act5C* knockdown flies with malformed footpads demonstrated decreased ability to adhere to a smooth substrate. Therefore, the formation of the functional footpad involves hair cell elongation, supported by a framework of actin filament bundles that provides a specific shape and facilitates cuticular deposition. This mechanism of formation of the microstructures of footpad hair inspired us to design a new adhesive device. We propose a new fiber-framed adhesive device with a spatula-like structure supported by nylon fibers with gel materials at the tip. This simple procedure, using a self-assembly mechanism, enables us to reduce the cost of raw materials and energy required to produce useful adhesion devices.

## Results

**Footpads in *Drosophila*.** In the fruit-fly, *Drosophila melanogaster*, the tarsus (distal part of the leg) is divided into five segments, and the pretarsus is the structure that forms the tip of the leg. A pair of claws and a pair of footpads are present on the dorsal and ventral sides of the pretarsus, respectively[9] (Fig. 1a). In addition, ~30 hairs (setae) are arranged in six or seven rows in the stalk of each footpad. The tip of each hair is ~1–2 μm in width, and has a spatulate shape (Fig. 1b). Cross-sectional images obtained using a transmission electron microscope (TEM) have shown that each hair is a flat tube composed of a cuticle (Fig. 1c). Flies secrete a viscous fluid between the footpad and the substrate and use attractive capillary forces to attach to the surface[6] (Supplementary Fig. 1).

**Footpad formation during metamorphosis.** To examine the processes involved in the formation of these footpads during the pupal stage in *Drosophila* development, we first screened suitable GAL4 strains for the capacity to label footpad cells with a GFP expression marker (Supplementary Fig. 2). We determined that we could effectively label most of the footpad hair cells and some of the cells associated with the ventral aspect of the claws, in a *seven-up* (*svp*)-GAL4 line marked with GFP (*svp > GFP*). Then the

*svp*-expressing cells were labeled with actin5C (*Act5C*: a gene encoding cytological actin in *Drosophila*)-GFP and nucleus-targeted with DsRed (*svp > Act5C-GFP, nucDsRed*) (Fig. 1d–g, d′–g′) and the formation of the footpad was monitored. At 20 h after puparium formation (AFP), we observed two clusters of GFP-expressing cells in ventral epidermal cells of the pupal pretarsus, which is formed from a simple sack of epidermal cells at this stage (Fig. 1d, d′). Each of these clusters of cells then began to extrude the processes outwards (Fig. 1e, e′). The cell bodies marked with magenta nuclei, in these footpad-forming cells, were seen to migrate proximally, while the tips remained in fixed positions (Fig. 1f, f′; Supplementary Movie 1). By 40 h APF, the cells had completed the construction of the framework for the footpad design (Fig. 1g, g′). Thus, the structural morphology of the *svp*-expressing cells was seen to dramatically change during the formation of the footpad during the first half of the pupal stage. Once the framework was completed, the process of depositing the cuticle began. To address whether a single cell forms a single hair or multiple hairs, we labeled only a limited number of *svp*-expressing cells, using the mosaic analysis with a repressible cell marker (MARCM) method[13]. Our results revealed that a single-labeled cell extended a fine process, which formed the tip of a single hair (Supplementary Fig. 3, Supplementary Movie 2), indicating that a single cell forms a single hair on the footpad. A similar mechanism was reported in the *Stenus* species[4], suggesting that this process is conserved across insect species.

**Cytoskeletal actin fiber involvement in footpad formation.** It is well documented that cytoskeletal actin plays a vital role in morphological changes of cells[14]. Therefore, we next examined the role of cytoskeletal actin in the formation of footpads. To accomplish this, we stained the actin filaments in hair cells, using fluorescent phalloidin, which binds to fibrous actin molecules. Actin filaments were found to accumulate in the apical aspect of cells in the footpads at 24 h APF (Fig. 2a). Following 30 h APF, the cells began to extend the processes containing actin filaments stained with phalloidin (Fig. 2b; Supplementary Fig. 4); and at 40 h APF, actin filaments were observed to assemble and form the framework for a spatulate shape within the cell. Cytoskeletal actin filaments were, therefore, determined to have a role in the formation of the spatulate shape of the individual hair cells of the footpads (Fig. 2c–c‴).

To further elucidate the role of actin filaments, we examined the effects of RNAi-mediated knockdown of cytoskeletal actin (*Act5C*) on footpad formation. *Act5C RNAi* was induced in hair cells under the control of an *svp*-GAL4 driver. To avoid potentially lethal effects, *Act5C RNAi* was produced following the creation of the puparium, using the temperature-sensitive suppressor Gal80[TS] of GAL4[15] (*svp > Act5C-RNAi, GAL80[TS]*, Fig. 2d). Alternatively, in another group of flies, the effects of *Act5C RNAi* were inhibited in the nervous system by induction of pan-neuronal expression of *elav-GAL80*[16] (*svp > Act5C-RNAi, elav-GAL80*, Fig. 2e). The flies survived to the adult stage of development, and thus, we could effectively observe the effect of *Act5C* knockdown on footpad formation. In both groups of flies, RNAi-mediated *Act5C* knockdown induced malformation of the hair shape, resulting in forked-shape tips rather than the normal spatulate shape (Fig. 2d–f). This abnormal shape should have been the result of the disrupted formation of the hair tip framework. No effects other than the malformed tips were seen in footpad morphology or the process of footpad formation in the *Act5C* knockdown flies under our experimental conditions (Supplementary Fig. 5, Supplementary Movie 3), probably because the level of knockdown was not high enough to induce other effects.

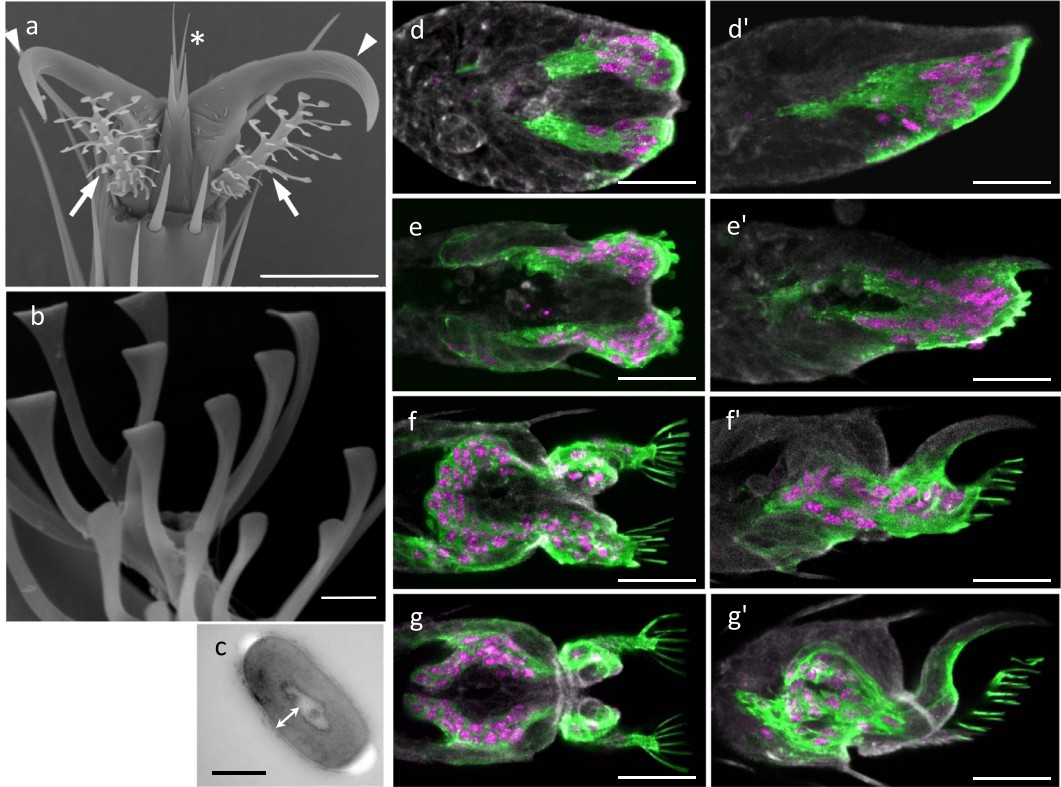

**Fig. 1 Formation of the footpad during fly metamorphosis. a** SEM image of footpads in wild-type Canton-Special (CS) fly. Arrows: pair of footpads; arrowheads: pair of claws; asterisk: empodium. Scale bar: 20 µm. **b** Hairs on a *Drosophila* footpad, depicting the clear spatulate-shaped structures. Scale bar: 2 µm. **c** A cross-sectional TEM image across the stalk of a footpad hair. Arrow: cuticle. Scale bar: 0.2 µm. **d–g, d'–g'** Development of footpad cells labeled with actin-GFP (green) and nucleus-targeting DsRed (magenta) in *svp > Act5C::GFP, nucDsRed*. Pretarsal cells are outlined by fluorescent phalloidin staining (gray). **d–g** Ventral view; **d'–g'** lateral view. The right side of the images shows the distal tips of the pretarsus. **d, d'** 20 h APF. **e, e'** 28 h APF. **f, f'** 36 h APF. **g, g'** 40 h APF. Scale bar: 20 µm.

We next examined whether *Act5C* knockdown flies (*svp > Act5C RNAi, elav-Gal80*) with malformed footpads demonstrated defects in climbing ability on smooth substrates (Fig. 3; Supplementary Movies 4 and 5). We found that it took *Act5C* knockdown flies twice as long to climb the vertical wall of a Pasteur pipette, compared with the control flies with normal footpads (Fig. 3a, c, d). Moreover, the *Act5C* knockdown flies were seen to lose traction and slip down the pipette wall. They also struggled to effectively pass the constriction point on the pipette, resulting in their falling from the wall, while normal flies managed this obstacle with ease (Fig. 3b–d). Hence, the malformed footpads should reduce the active contact area between the tip of each hair and the substrate, resulting in decreased ability to adhere to smooth substrates. Taken together, these results demonstrate that cytoskeletal actin filaments are a mandatory component in the mechanism employed by *Drosophila* for the formation of the structural framework that determines the shape and thereby adhesiveness of the footpad hair. The footpad hair tips in insects show various shapes, such as filamentous, lanceolate, spatulate, and discoidal[3–5,17]. Our results suggest that insects with hairy pads employ this actin-based morphogenesis mechanism to provide distinct shapes to the microstructures of their footpad hair.

**Biomimetic adhesive device inspired by the footpad formation.** This mechanism employed by *Drosophila* in the formation of footpad microstructures provided insights into the designing of a new biomimetic device to improve adherence to substrates. Generally, the processes responsible for the establishment of the footpad microstructures in a fly are divided into two steps (Fig. 4a). First, a spatulate-shaped framework is formed by actin fibers at the tip of the hair. Secondly, cuticle materials are secreted, accumulated, and then solidified. We, therefore, propose a strategy for manufacturing adhesive structures using a similar two-step system (Fig. 4b). Step 1: design the structural framework with nylon fiber. Step 2: immerse the framework in resin and allow the spatulate-shaped structure to form via surface tension. The resulting adhesive structure contains a spatulate structure supported by nylon fibers, with a gel calcium alginate material forming the tip, which is the point of contact for a substrate (Fig. 4c). We evaluated the adhesive properties of the hair-like structures by performing a shear test. Since the setae of the fly are covered with a secreted fluid, we carried out the test after dipping our artificial hair structures in water (Fig. 4d). It was shown both in beetles[18] and mimicked artificial systems[19] that the amount of liquid at the hair tip could strongly influence the adhesion forces. This is the case in our devices also. The adhesion forces in the fibrillar system decreased with increasing amounts of water and increased with decreasing amounts of water. Immediately after dipping in water, the adhesive device had a high water content, and adhesive strength of $40 \pm 15$ mN (mean ± SE, $n = 6$). However, when the structure was almost dried after contact, the device exhibited a high average adhesive strength ($779 \pm 108$ mN, mean ± SE, $n = 11$), which is equivalent to the force capable of suspending a 60-kg human being with an adhesive area of 9 cm² (comparable with ~756 fibers), or a circle with a radius of ~17 mm. Figure 4e and f demonstrates that a single adhesive

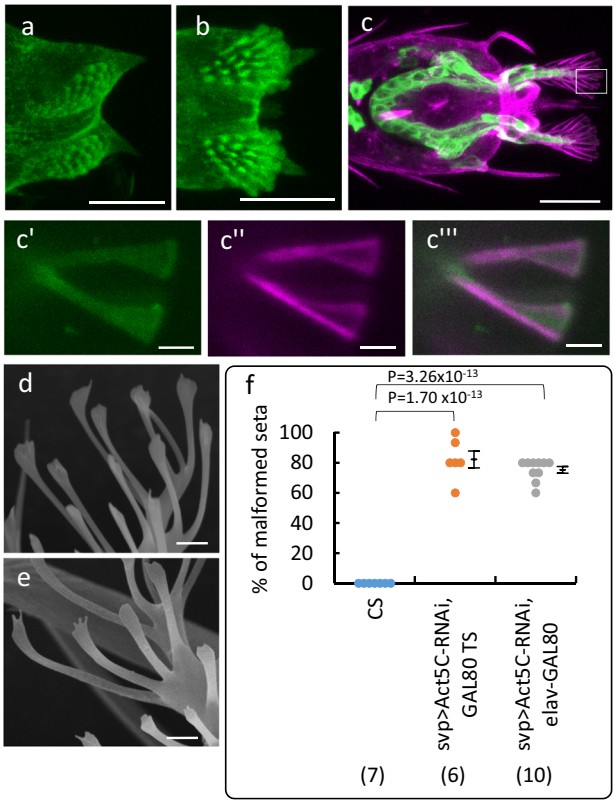

**Fig. 2 Role of cytoskeletal actin in the formation of fly footpads.**
**a**, **b** FITC-conjugated phalloidin-stained pretarsi of Canton-Special (CS) pupa at 24 h (**a**) and 30 h APF (**b**), showing the accumulation of cytological actin in the footpad hair cells. **c-c‴** TexasRed-conjugated phalloidin staining of the pretarsus at 40 h APF in *svp > mCD8::GFP* flies. The boxed region in panel **c** is enlarged in **c′–c‴**, highlighting the tip of a hair cell. **c′** mCD8::GFP (green). **c″** TexasRed staining (magenta). **c‴** A merged image of GFP and TexasRed staining. The right side of images in (**a–c‴**) shows the distal tips of the pretarsus. **d**, **e** SEM images of malformed footpad hairs in *Act5C* knockdown flies of *svp > Act5C RNAi, TubP-GAL80ᵀˢ* and of *svp > Act5C RNAi, elav-Gal80* flies, respectively. **f** Effect of *Act5C* knockdown on the formation in hair tips. The graph represents the percentage of malformed (forked) hairs (dot plots on the left and the average ± SE on the right in each column). The data were analyzed using the Tukey–Kramer test for all pairwise comparisons: *P*-values are shown on the top. The flies used were of *CS strain* (wild-type), *svp > Act5C RNAi, TubP-GAL80ᵀˢ* and of *svp > Act5C RNAi, elav-Gal80* flies. The number of flies examined is shown in parentheses on the *x* axis. Scale bars: 20 μm in (**a–c**) and 2 μm in (**c′–c‴**, **d**, **e**).

hair is capable of suspending a 52.8-g silicon board. Friction and adhesion of single pads were measured in beetles[20], which suggested that the distal pads containing spatula-tipped hairs, for the most part, are mainly used for pulling and adhesion. So, comparison with the value of the shear stress of distal pads on a smooth surface showed that the shear stress of our device, 784 ± 144 kPa, is comparable with that of distal pads on a smooth surface, 642 ± 99 kPa in males and 613 ± 111 in females. Since our device is weak against peeling in a perpendicular direction, this adhesive ability is easily reversed by applying an opposite directional force or by simply twisting the fiber, similar to what is observed in insect footpads[20–22]. Importantly, this apparatus maintains its adhesive properties after repeated attachment and detachment cycles, even though it is necessary to dip it again in water before use when it dries up completely.

The attached area of our adhesive structure (1.2 mm² ± 0.1, mean ± SE, *n* = 11) was about three orders of magnitude larger than that of the fly's footpad hair (3 μm² ± 0.1, mean ± SE, *n* = 5). Here, the adhesion mechanism in relation to the effect of size should be considered. In general, as the size of an object decreases, the object becomes affected more by the surface than the volume. Therefore, a small object can be lifted by the surface tension of a small droplet. In the case of the fly's hair and the developed adhesive structure, the surface is larger than the volume in both cases because of the film shape. The mechanism of adhesion of such a structure via a low-viscosity liquid changes, depending on the amount of the liquid. Such a phenomenon is the same for living and artificial things. When the liquid volume is large, the distance between the adhesive and the adherend is large, and the capillary force is the main force responsible for adhesion. As the liquid volume decreases and the thickness of the interposed droplets become smaller than the radius of the contact portion, the Laplace pressure is responsible for adhesion. Further, when the amount of the droplet is small, and the solid surface is very close, the intermolecular forces come in play. When the contact area is the same, the adhesive force is smaller when the liquid amount is higher and vice versa. Therefore the adhesive force is determined by the surface to volume ratio, the amount of liquid present, and several interactive forces.

**Conclusion.** By examining the mechanism responsible for the development of footpad microstructures in flies, we biomimetically designed an effective adhesive agent. Since 2005, the development of biomimetic adhesive devices has been based on studies on biological dry hair adhesives, such as those found on geckos[23–26]. However, adhesion on a rough surface limits the true contact area between the adhesive structure and the deposition surface. Thus, to establish a larger contact area to improve the effectiveness of the adhesive surface, it is mandatory to incorporate flexible contact-forming hairs in the design. It is known that the stiffness of insect hair is not uniform, for example, a material composition gradient and subsequent Young's modulus gradient in the longitudinal direction of the hairs is observed as in the ladybird beetle footpad[27]. The base of hair stalk is sufficiently hard to prevent clustering, and the contact surface of hair tips is sufficiently compliant to adapt to rough substrates. In addition, the bridge between the hair tip and the substrate is filled with the secreted liquid, which generates the capillary forces that adhere the insect to the substrate[6]. In the adhesive device, as in the biological setae, softness at the tip is important for adhesion. Through the use of a material with high moisture content, we designed highly flexible thin structures located at the device tip, thereby increasing the real contact area and improving the adhesive ability on rough surfaces. Furthermore, it is vital to incorporate hair-like fibers with high tensile strength capable of withstanding repeated attachments and detachments, which was achieved through the use of nylon fibers in our device. The compliance of the structure is known to strongly influence the adhesion force[19]. Further validation of our adhesive device, including quantification of the compliance, should be carried to compare the adhesiveness of the device with that of insect footpads. The use of an uncured liquid raw material such as sodium alginate in our design would serve to simplify the manufacturing process and reduce the cost of production associated with our device. Our approach effectively applied mechanisms employed in developmental biology in a mimetic approach (developmental biomimetics) for the design of new biology-inspired devices. We have, therefore, demonstrated developmental biomimetics as a strategy for developing new concepts in material science.

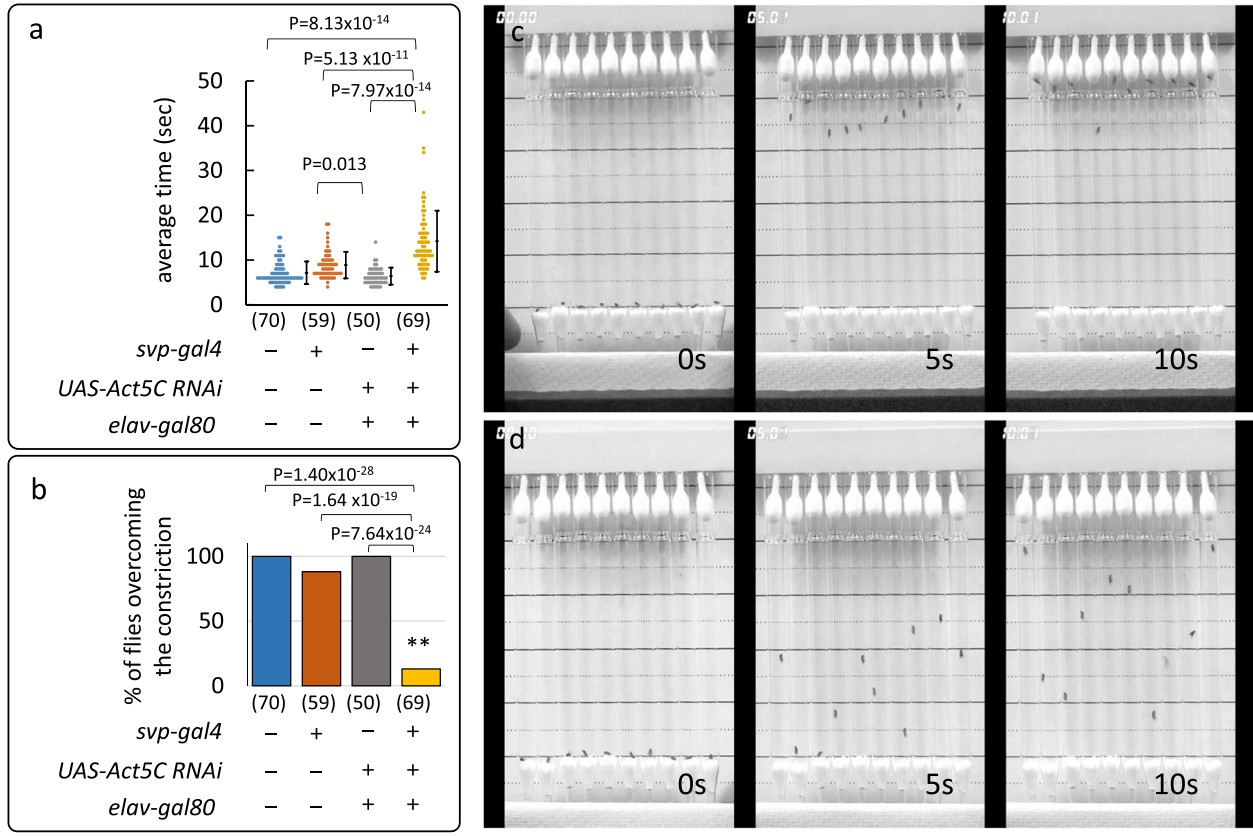

**Fig. 3 Decrease in climbing ability in the *Act5C* knockdown fly. a** Effects of *Act5C* knockdown on the climbing ability of flies on smooth substrates. The graph illustrates the time taken to climb the vertical glass wall of a Pasteur pipette from the bottom to the constriction (8 cm) (dot plots on the left and the average times (sec) ± SD on the right in each column). The data were analyzed using the Tukey–Kramer test for all pairwise comparisons: *P*-values are shown on the top. **b** The panel depicts the percentage of flies capable of overcoming the constriction point on the Pasteur pipette in 1 min. Kruskal–Wallis test, Scheff's post hoc: **\*\****P* < 0.001 (*P*-values are shown on the top). The flies used were of genotypes +; + (CS strain), +/+; *svp-GAL4/+*, *UAS-Act5C RNAi/+; elav-GAL80/+* and *UAS-Act5C RNAi/+; svp-GAL4/elav-GAL80*. The number of flies examined is shown in parentheses on the x axis. **c**, **d** Snapshots of wild-type CS and *Act5C* knockdown flies (*svp > Act5C RNAi, elav-GAL80*) climbing the walls of Pasteur pipettes, respectively.

## Methods

**Insects**. Fruit-flies, *Drosophila melanogaster*, were reared on cornmeal–yeast medium at 25 °C with constant illumination. CS flies were used as wild-type controls. For staging during metamorphosis, white prepupae were collected within 30 min after puparium formation (APF) and incubated at 25 °C until appropriate developmental stages were observed.

To determine which flies were suitable for labeling footpad cells, we crossed several GAL4 strains involved in pretarsus formation with UAS-mCD8::GFP strains and examined the expression pattern of GFP in the pretarsus region of the F1 pupal legs ~2 days APF. We examined *bar-, ckm10-, spa-, neur-, dpp-,* and *svp-GAL4 Drosophila* lines. *Bar-GAL4* and *dpp-GAL4* lines were generously donated by Dr. Kojima[12] and the *ckm10- GAL4* lines by Dr. C. Mirth[28]. The *spa-, neur-* and *svp-GAL4* lines were obtained from the Kyoto Stock Center. Strains of *FRTG13, UAS-mCD8::GFP, FRTG13, TubP-GAL80, UAS-nucDsRed,* and *TubP-Gal80*^TS *(7019)* were obtained from BDSC, and *UAS-Act5C::GFP* was obtained from the Kyoto Stock Center. A line of *UAS-Act5C RNAi (7139)* was obtained from VDRC. A line of *elav-Gal80* was gifted by S. F. Goodwin.

Detailed genotypes and the abbreviations of all strains used in the paper are indicated in Supplementary Table 1.

**Observation of footpad morphology**. The footpad morphology was examined using a scanning electron microscope (SEM). Fly legs were fixed with 3.7% for-maldehyde, washed in distilled water, and serially dehydrated with ethanol. After dehydration, ethanol was replaced with tert-butyl alcohol. The samples were then frozen at 4 °C and further dehydrated by freeze–drying (VFD-21S, Vacuum Device Inc). Specimens were then subjected to gold coating by an ion coater (Neo coater MP-19010 NCTR, JEOL), and observed with SEM (TM3000 Miniscope, Hitachi*).*

Legs were dissected from the body using fine scissors and prefixed overnight in 2% glutaraldehyde and 2% paraformaldehyde in 0.1 M cacodylate buffer (pH 7.2), following three washes for 10 min each in 0.1 M cacodylate buffer. Then, the specimens were postfixed for 2 h in 1% OsO₄ in 0.1 M cacodylate buffer at room temperature. Dehydration through a graded ethanol series and substitution for

propylene oxide were followed by embedding in Araldite and Quetol 812 resin (Nissin EM, Japan) mixture. Ultrathin sections were cut with an ultramicrotome (UCT; Leica, Germany) and stained with 2% uranyl acetate for 5 min and then in a lead-staining solution (Sigma-Aldrich, USA) for 3 min. The sections were observed by TEM (JEM-1220; JEOL, Japan), and the digital images were obtained with an attached cooled CCD camera (Gatan, USA).

**Observation of footpad formation during metamorphosis**. We employed *svp*^NP5606-*GAL4* and *svp*^NP0724-*GAL4* lines to assess the development of footpads in *Drosophila*, and since both strains demonstrated similar expression patterns of GAL4 in the footpad hair cells, we designated both as *svp-Gal4*. Furthermore, the hair cells in the footpads of pupae with the genotype *spa > mCD8::GFP, nucDsRed,* or *svp > Act5C::GFP, nucDsRed* were labeled with mCD8::GFP or actin5C-GFP or nucleus-targeted DsRed, and were dissected at appropriate developmental stages. Briefly, the legs were dissected, and the pupal cuticles were removed. The legs were then fixed with 3.7% formaldehyde for 30 min and washed with phosphate-buffered saline (PBS) containing 0.3% Triton-X (PBS-Tx). For immunochemical analysis, specimen preparations were incubated with anti-GFP primary antibodies (1:500, rabbit polyclonal; A6455, Molecular Probes, Eugene, OR) for over 6 h at 25 °C. After washing, the tissues were treated with Cy2-conjugated goat anti-rabbit IgG secondary antibodies (1:500, Jackson Immuno-Research) for over 6 h at 25 °C.

To visualize cytoskeletal actin filaments, the specimens were stained with TexasRed-X phalloidin (Invitrogen, T7471), Acti-stain^TM 555 phalloidin (Cytoskeleton, Inc. PHDH1-A), and Alexa Fluor^TM 633 (Invitrogen, A22284) for over 2 h. Images of the footpad were obtained with a Leica TCS SPE confocal microscope (Wetzlar, Germany) using Leica application suite advanced fluorescence (LAS AF) software.

Somatic clones were produced using the mosaic analysis with a repressible cell marker (MARCM) method, which allows for single-cell labeling by inducing chromosomal recombination, enabling marker expression only in rare clones. White pupae of the *y hs-flp; FRTG13, UAS-mCD8::GFP/ FRTG13, TubP-GAL80; svp-GAL4/+* genotypes were collected and maintained for 6 h. They were then subjected to a heat-shock at 37 °C for 30 min. At 42 h APF, the pupae were

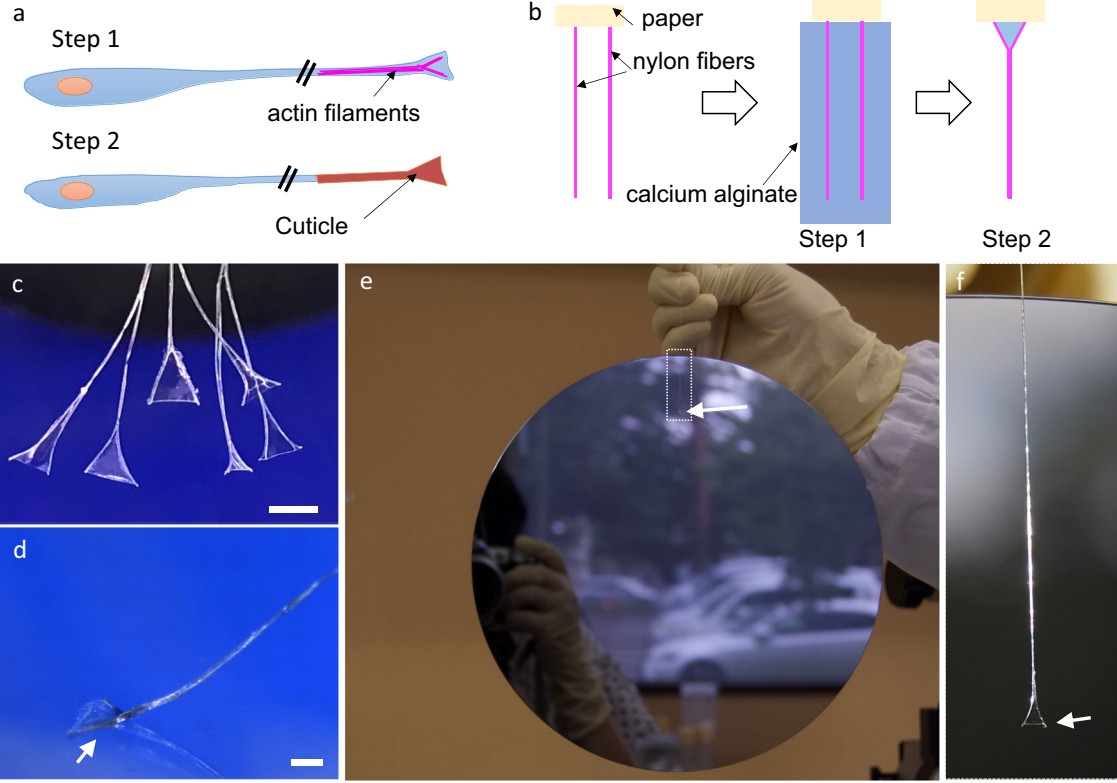

**Fig. 4 Design of a biomimetic adhesive device inspired by the mechanism of fly footpad formation. a** Schematic diagrams for the formation of the footpad hair in flies. In the first step, at the tip of a hair, a framework for the spatulate structure is created by actin filaments. In the second step, the deposition of cuticle occurs on the framework. **b** Schematic diagrams for the formation of a biomimetic hair-like structure (fiber-framed adhesive structure). Step 1: a framework is created with nylon fibers. Step 2: the framework is immersed in resin and self-assembles into a spatula-shaped structure. **c** Biomimetic fibers are seen to frame the adhesive structures. Scale bar: 500 μm. **d** A fiber is attached to the glass substrate with liquids (arrow). Scale bar: 500 μm. **e** A 52.8-g silicon board (diameter = 8 inch) is suspended by a single biomimetic fiber. Arrow indicates the attachment site. **f** The attachment site of the adhesive fiber (arrow). The boxed region in (**e**) is enlarged.

dissected, and the legs were fixed with 10% formalin for 30 min and stained for immunohistochemistry analysis as described above.

Live images of footpad development were obtained during the pupal stage in *svp > mCD8::GFP, nucDsRed, svp > Act5C::GFP, nucDsRed, or svp > mCD8::GFP, Act5C RNAi, elav-Gal80* flies. At 20 h APF, following the removal of the puparium, pupae were attached to a glass coverslip. The coverslip was then glued to a tissue culture slide (Lab-Tek), with the pupae facing downwards, with wax, leaving a small space next to the coverslip to avoid condensation. To prevent the preparation from drying, distilled water was added to a chamber in the tissue culture slide. Time-lapse images for GFP and DsRed-labeled cells were obtained every 20 min using the Leica TCS SPE confocal microscope. Under these experimental conditions, pupae that are lacking puparium will develop into adults.

**RNAi-mediated knockdown of cytoskeletal actin in flies**. To examine the effects of RNAi-mediated knockdown of cytoskeletal actin (*Actin5C*) on footpad formation in flies, *Act5C RNAi* was introduced into hair cells under the control of an *svp-GAL4* driver. To avoid lethal effects in *svp-Gal4 > Act5C RNAi* flies, *Act5C RNAi* was introduced by one of the two methods. The first involves treatment with the RNAi after the formation of the puparium, using a temperature-sensitive suppressor Gal80$^{TS}$ of GAL4. The flies of *svp > Act5C RNAi, TubP-GAL80$^{TS}$* were reared at 18 °C during the larval stages. The prepupae obtained at 0–12 h APF were then transferred to 30 °C and maintained at a controlled temperature. Using these conditions, some adult flies were enclosed, and the morphology of their footpads was examined by SEM. The second method of introduction protects against the adverse effects of *Act5C RNAi* on the nervous system using elav-GAL80. Using this method, the flies of *svp > Act5C RNAi, elav-Gal80* survived to adulthood, and we were able to observe the effects of *Act5C* knockdown on footpad formation. We counted the number of hairs with malformed tips among 15 hairs from three distal rows in one stalk of each footpad.

**Climbing test**. To assess the ability of knockdown flies to climb effectively, adults were collected within 24 h of being enclosed and maintained in vials containing Kimwipe paper soaked in 20 mM sucrose solution. Each female fly, at 24–48 h after having been enclosed, was placed in a Pasteur pipette, both sides of which were

then plugged with cotton. The fly was gently tapped down to the bottom of the pipette and assessed for its climbing ability using a video recorder (Evrio JVC, Japan) for 1 min. The time required to climb the vertical glass wall of a Pasteur pipette from the bottom to its constriction site (8 cm) was recorded. In addition, the number of flies capable of overcoming the constriction point of the Pasteur pipette within 1 min was determined. Data from flies that were unable to reach the constriction point within 1 min were discarded to calculate the average times. The flies used were of genotypes+: + (CS strain), +/+; *svp-GAL4/+, UAS-Act5C RNAi/+; elav-Gal80/+* as controls and *svp > Act5C RNAi, elav-Gal80* for experimental strains.

**Manufacturing the biomimetic fiber-framed adhesive structure**. Nylon fibers with a diameter of 52 μm, sodium alginate solution (1 wt%), and calcium lactate solution (1 wt%) were prepared for the fabrication of the spatulate structure. At first, two nylon fibers at 2-mm intervals were fixed on a paper at one extremity. The fibers were then dipped in a sodium alginate solution and then lifted up from it. When the fibers were removed from the sodium alginate, a spatulate structure was formed owing to the surface tension of the solution[29,30]. The structures were then placed in a calcium lactate solution, where they solidified. After air-drying, the spatula was cut out from the paper substrate. The hair-like structures were hydrated prior to use. The area of the spatulate structure was 1.2 mm$^2$ ± 0.1 (mean ± SE, $n = 11$). The whole process was carried out at 23 °C.

**Shear force measurement**. Adhesive testing of fly-mimetic hair structures was performed on a glass substrate at 23 °C using a load cell force transducer with a 980 mN capacity (World Precision Instruments, Sarasota, FL, USA). Before performing the assay, the glass substrate was cleaned in an ultrasonic bath with acetone, ethanol, and distilled water for 10 min. The test was carried out after dipping our artificial hair structures in water. The amount of water loaded on the structure during dipping was 560 ± 5 μg (mean ± SE, $n = 6$). Shear force measurements were performed by moving the glass substrate at a lateral speed of 24 mm/min.

**Statistics and reproducibility**. Statistical tests were performed in Microsoft Excel using BellCurve for Excel (version 3.20) software. The appearance of malformed

(forked) hairs in *Act5C* knockdown flies was compared to that in the control in *CS* flies, using one-way ANOVA followed by Tukey–Kramer test. *P*-values are shown on the top of the graph. The evaluation of statistical significance of differences of the average times taken to climb the vertical glass wall was performed with one-way ANOVA and Tukey–Kramer test to compare wild-type and *Act5C* knockdown flies. Climbing assay to compare the percentage of flies capable of overcoming the constriction was analyzed using Kruskal–Wallis test, and post hoc pairwise comparisons were conducted using Scheff's post hoc test. Measurements were acquired from distinct samples. Sample sizes for each experiment and *P*-values obtained from individual statistical analyses are indicated in the figures. The reproducibility in experimental qualitative data such as immunostaining or fluorescence-staining was verified by repeating the analyses in at least five biological samples, independently. All replication attempts were successful. Sample sizes for each measurement experiment concerning the biomimetic fiber-framed adhesive structure are also described in the texts and figure legends.

**Reporting summary**. Further information on research design is available in the Nature Research Reporting Summary linked to this article.

## Data availability

The data that support the findings of this study are available from the corresponding author upon reasonable request. Source data underlying plots are provided in Supplementary Data 1.

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

## Acknowledgements

We would like to thank T. Kojima, C. Mirth, S.F. Goodwin, the Kyoto Stock Center, VDRC (Vienna *Drosophila* Resource Center), and BDSC (Bloomington *Drosophila* Stock Center) for the various *Drosophila* strains. We would also like to thank A. Izumi, C. Sato, Y. Koizumi (HUE), H. Niu (National Institute for Materials Science: NIMS), and K. Koitabashi (NIMS) for their technical assistance. This work was supported by a Grant-in-Aid for Scientific Research on Innovative Areas from the Ministry of Education, Culture, Sports, Science, and Technology (MEXT) of Japan to K.-i.K. and T.H. (No. 24120004) and by "Seeds" Development Research Grants of NIMS to N.H.

## Author contributions

K.-i.K., R.M., Y.Y., and N.H. performed the experiments; K.-i.K., R.M., Y.Y., T.H., and N.H. designed and interpreted the experiments; K.-i.K. and N.H. wrote the paper.

## Competing interests

The authors declare no competing interests.
