## [Peer Review File · Communications Biology]

Reviewers' comments:

Reviewer #1 (Remarks to the Author):

This is an interdisciplinary study integrating insect biology and biomimetics of surface engineering / science, which addresses the role of framework with cytoskeletal actin filaments for insect's footpad formation. The authors successfully visualized the developmental formation of the footpad hairs in *Drosophila melanogaster* and revealed that the elongation of the hair cells and assembly of actin filaments play a crucial role in forming effective frameworks with actin filament bundles. To assess the adhesive function of the framework – a spatula-like structure, the authors further proposed a biomimetic design of adhesive devices by means of a self-assembly mechanism. I think that this study could be a good contribution to the literature body of biomimetics in insect-inspired engineering if the authors could clarify the following two major issues.

1) Both biological and biomimetic parts are very impressive but independently and there exists an obvious unreasonable gap in bridging the two, namely, the size or scaling effect. The representative scale or size of the spatulate tip in a single hair of *Drosophila melanogaster* is on the order of 20 μm (Figs. 1-2), where the Van der Waals, Coulomb, and attractive capillary forces may work together because hairs are clustered to bundles with a secreted fluid among them. On the other hand, it is aware that the scale of the artificial spatulate tip of a single hair-like structure is of 500 μm (Fig. 3), 25 times the biological ones on the footpads, and it seems that the surface tension actually plays a crucial role at such a scale. Thus, the biological design and the biomimetic design very likely do not share the same mechanism in terms of adhesive functions. This may be just a morphologically inspired biomimetic adhesive device rather than a functional one. I would suggest that the authors clarify the issue probably from the viewpoint of scaling or size effects on the microscopic forces.

2) In addition, if the framework design would play a crucial role in enhancing the adhesive capability, the morphological effects could be of great importance and may work differently at different scales of 25 or 500 μm . An extensive study or discussion needs to offer an investigation on this issue as well.

Reviewer #2 (Remarks to the Author):

This manuscript presents an experimental investigation of the formation of hairy adhesive pads in flies, as well as a biomimetic process to generate upscaled artificial adhesive structures. Firstly, the authors show that spatulate-shaped setae are formed through the assembly of actin filaments that serve as a backbone for the further deposition of cuticular material. They convincingly prove that actin filaments are responsible for the particular shape of the seta tips and the resulting efficient adhesion. Indeed, when the Act5C gene is knocked down, the hair tips are badly shaped, and the authors prove that it significantly diminishes the locomotion ability of flies on smooth surfaces. In a second time, the authors propose a novel fabrication technique of a millimetre-scale adhesive pad inspired from the biological shaping process. They demonstrate its ability to maintain adhesion under high shear forces.

The paper reports an original investigation of a biological mechanism that they successfully transpose to engineering. This study will certainly be of interest to the readers of *Communications Biology*, as well as to engineers interested in mimicking the shaping mechanism. I would recommend its publication when the following comments are addressed:

1. Line 47: This simple procedure, using a self-assembly mechanism, enables us to reduce the cost and energy requirement. The cost and energy requirement of what? This should be clarified.

2. Line 87: indicating that a single cell forms a single hair on the footpad. A similar conclusion was reached by O. Betz for the *Stenus* species (cf. Betz, *J. Morphol.* 255, 24-43, 2003). Maybe this similarity across species is worth mentioning. More generally, could the authors shortly discuss the extent to which this actin-based morphogenesis mechanism is likely to be observed for other insects with hairy pads (e.g. beetles, or other fly species)?
3. Line 132: after dipping our artificial hair structures once in the water. How much water is loaded on the structure during dipping? Is this amount reproducible? It was shown both on beetles (e.g. Bullock et al., *J. Exp. Biol.* 211, 3333-3343, 2008) and mimicked artificial systems (e.g. Gilet et al., *Soft Matter* 15, 3999, 2019) that the amount of liquid at the seta tip could strongly influence the adhesion forces (both normal and tangent to the substrate).
4. Line 139: Importantly, this apparatus maintains its adhesive properties after repeated attachment and detachment cycles. Is there any dipping in water needed between two successive attachments?
5. Line 147: we designed highly flexible thin structures located at the device tip Nothing is said about the resulting compliance of these structures (which would depend on both dimensions and Young's modulus of each material). However, this compliance is known to strongly influence the adhesion force (Gilet et al., *Soft Matter* 15, 3999, 2019). Therefore, it should be quantified and the corresponding elastocapillary number should be compared to that of either fly or beetle seta tips.
6. Line 273: A cross-sectional TEM image of a footpad hair The authors should specify the approximate location of the cross-section: is it across the seta tip or across the stalk?
7. Line 414: The ends of the fibres were fixed to the supporter at 2 mm intervals. What was the relative orientation of the fibres before immersion? Was this orientation locally maintained after immersion? Were the fibres clamped at both extremities or only at one?
8. Line 416: a spatulate structure was formed by the surface tension of the solution It would be nice to refer to the existing literature on fibres deformed by surface tension, e.g. Duprat and Protiere, *EPL* 111, 56006 (2015) or Duprat et al., *Nature* 482, 510 (2012).
9. Line 421: Adhesive testing of fly-mimetic hair structures was performed Is it normal adhesion or shear adhesion? In figure 3e, it is obviously a shear load at play (i.e., tangent to the contact area). Has the normal adhesion be tested ? How do the shear stress and the shear force per unit width compare to results obtained for similar experiments on beetles? (e.g. in Bullock and Federle, *J. Exp. Biol.* 212, 1876, 2009)
10. There are several spelling / grammar mistakes, including in the abstract. Some careful proofreading is needed

Reviewer #3 (Remarks to the Author):

The authors highlight the role of actin in the formation of shape and structure of setae with a spatula shaped end. This shape of foot hairs are of interest from many years due to the interest in adhesion of insects and geckos to smooth vertical surfaces. This has also prompted a significant interest in developing synthetic structures for robotics and other applications.

The authors have provided some very intriguing results of the formation of setae and with the staining

we can learn about the formation of the setae and perhaps the shape of the tip.

I want the authors to address these questions and make changes in the manuscript where applicable.

1. Provide some quantitative analysis of the shape of the spatula and how some statistics on how many spatula were malformed due to Act5C knock-out.
2. Elaborate what other functions have been affected by Act5C knock-out other than just the malformed spatula. Based on the movies on how the structure is formed, it is even surprising that even the setae are formed when you do the Act5C knock-out. It appears from the evidence provided that the actin filaments could be also playing an important role in retraction of cells since the tips are remaining in the same place.
3. Have the authors stained for lipids? In the gecko literature there is evidence for lipids covering the setae (Sci. Rep. 2015, 5, 9594).
4. Can the authors comment on the presence of keratin forming proteins.
5. The authors mention in many places that the tips are softer. There is no mechanical data provided and this has to be corrected.
6. The analysis of synthetic structures are very preliminary. No discussion is provided on what role the shape plays. What is the role of capillary forces. More discussion needs to be provided.

Reviewer #1 (Remarks to the Author):

This is an interdisciplinary study integrating insect biology and biomimetics of surface engineering / science, which addresses the role of framework with cytoskeletal actin filaments for insect' s footpad formation. The authors successfully visualized the developmental formation of the footpad hairs in *Drosophila melanogaster* and revealed that the elongation of the hair cells and assembly of actin filaments play a crucial role in forming effective frameworks with actin filament bundles. To assess the adhesive function of the framework – a spatula-like structure, the authors further proposed a biomimetic design of adhesive devices by means of a self-assembly mechanism. I think that this study could be a good contribution to the literature body of biomimetics in insect-inspired engineering if the authors could clarify the following two major issues.

1) Both biological and biomimetic parts are very impressive but independently and there exists an obvious unreasonable gap in bridging the two, namely, the size or scaling effect. The representative scale or size of the spatulate tip in a single hair of *Drosophila melanogaster* is on the order of $20 \mu\text{m}$ (Figs. 1-2), where the Van der Waals, Coulomb, and attractive capillary forces may work together because hairs are clustered to bundles with a secreted fluid among them. On the other hand, it is aware that the scale of the artificial spatulate tip of a single hair-like structure is of $500 \mu\text{m}$ (Fig. 3), 25 times the biological ones on the footpads, and it seems that the surface tension actually plays a crucial role at such a scale. Thus, the biological design and the biomimetic design very likely do not share the same mechanism in terms of adhesive functions. This may be just a morphologically inspired biomimetic adhesive devise rather than a functional one. I would suggest that the authors clarify the issue probably from the viewpoint of scaling or size effects on the microscopic forces.

2) In addition, if the framework design would play a crucial role in enhancing the adhesive capability, the morphological effects could be of great importance and may work differently at different scales of 25 or $500 \mu\text{m}$. An extensive study or discussion needs to offer an investigation on this issue as well.

To address Reviewer #1's comments 1) and 2), we added the following text about size effect and adhesion mechanism in the discussion.

The attached area of our adhesive structure ($1.2 \text{ mm}^2 \pm 0.1$, mean \pm SE, $n=11$) was about three orders of magnitude larger than that of the fly's footpad hair ($3 \text{ }\mu\text{m}^2 \pm 0.1$, mean \pm SE, $n=5$). Here, the adhesion mechanism in relation to the effect of size should be considered. In general, as the size of an object decreases, the object becomes affected more by the surface than the volume. Therefore, a small object can be lifted by the surface tension of a small droplet. In the case of the fly's hair and the developed adhesive structure, the surface is larger than the volume in both cases because of the film shape. The mechanism of adhesion of such a structure via a low-viscosity liquid changes depending on the amount of the liquid. Such a phenomenon is the same for living and artificial things. When the liquid volume is large, the distance between the adhesive and the adherend is large, and the capillary force is the main force responsible for adhesion. As the liquid volume decreases and the thickness of the interposed droplets become smaller than the radius of the contact portion, the Laplace pressure is responsible for adhesion. Further, when the amount of the droplet is small, and the solid surface is very close, the intermolecular forces come in play. When the contact area is the same, the adhesive force is smaller when the liquid amount is higher and vice versa. Therefore the adhesive force is determined by the surface to volume ratio, the amount of liquid present, and several interactive forces.

Reviewer #2 (Remarks to the Author):

This manuscript presents an experimental investigation of the formation of hairy adhesive pads in flies, as well as a biomimetic process to generate upscaled artificial adhesive structures. Firstly, the authors show that spatulate-shaped setae are formed through the assembly of actin filaments that serve as a backbone for the further deposition of cuticular material. They convincingly prove that actin filaments are responsible for the particular shape of the seta tips and the resulting efficient adhesion. Indeed, when the Act5C gene is knocked down, the hair tips are badly shaped, and the authors prove that it significantly diminishes the locomotion ability of flies on smooth surfaces. In a second time, the authors propose a novel fabrication technique of a millimetre-scale adhesive pad inspired from the biological shaping process. They demonstrate its ability to maintain adhesion under high shear forces.

The paper reports an original investigation of a biological mechanism that they successfully transpose to engineering. This study will certainly be of interest to the readers of Communications Biology, as well as to engineers interested in mimicking the shaping mechanism. I would recommend its publication when the following comments are addressed:

1. Line 47: This simple procedure, using a self-assembly mechanism, enables us to reduce the cost and energy requirement. The cost and energy requirement of what? This should be clarified.

This sentence was changed to ‘This simple self-assembly mechanism facilitates the energy-efficient production of low-cost adhesion devices.’

2. Line 87: indicating that a single cell forms a single hair on the footpad. A similar conclusion was reached by O. Betz for the *Stenus* species (cf. Betz, *J. Morphol.* 255, 24–43, 2003). Maybe this similarity across species is worth mentioning. More generally, could the authors shortly discuss the extent to which this actin-based morphogenesis mechanism is likely to be observed for other insects with hairy pads (e.g. beetles, or other fly species)?

Based on the reviewer’s suggestion, we have revised this section in the manuscript as follows.

“A similar mechanism was reported in the *Stenus* species, suggesting that this process is conserved across insect species.” (see, line 103-104)

“The footpad hair tips in insects show various shapes, such as filamentous, lanceolate, spatulate, and discoidal. Our results suggest that insects with hairy pads employ this actin-based morphogenesis mechanism to provide distinct shapes to the microstructures of their footpad hair.” (see, line 143-146)

3. Line 132: after dipping our artificial hair structures once in the water. How much water is loaded on the structure during dipping? Is this amount reproducible? It was shown both on beetles (e.g. Bullock et al., *J. Exp. Biol.* 211, 3333–3343, 2008) and mimicked artificial systems (e.g. Gilet et al., *Soft Matter* 15, 3999, 2019) that the amount of liquid at the seta tip could strongly influence the adhesion forces (both normal and tangent to the substrate).

We measured the amount of water loaded on the structure during dipping and have added this information in the Methods section. (see, line 343-345)

“The test was carried out after dipping our artificial hair structures in water. The amount of water loaded on the structure during dipping was $560 \pm 5 \mu\text{g}$ (mean \pm SE, n=6).”

Also, we agree with the reviewer's comment that the adhesive strength of each adhesive structure depends on the amount of water. So we added the following sentence in the revised manuscript. (see, line 160-169)

It was shown both in beetles and mimicked artificial systems that the amount of liquid at the hair tip could strongly influence the adhesion forces. This is the case in our devices also. The adhesion forces in the fibrillar system decreased with increasing amounts of water and increased with decreasing amounts of water. Immediately after dipping in water, the adhesive device had a high water content, and adhesive strength of 40 ± 15 mN (mean \pm SE, n=6). However, when the structure was almost dried after contact, the device exhibited a high average adhesive strength (779 ± 108 mN, mean \pm SE, n=11), which is equivalent to the force capable of suspending a 60 kg human being with an adhesive area of 9 cm² (comparable to approximately 756 fibers), or a circle with a radius of approximately 17 mm. Fig. 4e demonstrates that a single adhesive hair is capable of suspending a 52.8 g silicon board.

4. Line 139: Importantly, this apparatus maintains its adhesive properties after repeated attachment and detachment cycles. Is there any dipping in water needed between two successive attachments?

Adhesion is possible if the apparatus was sufficiently wet. When it dries up completely, it is necessary to dip it again in water before use. So, we added the following sentence to the discussion (line 176-178).

“Importantly, this apparatus maintains its adhesive properties after repeated attachment and detachment cycles, even though it is necessary to dip it again in water before use when it dries up completely.”

5. Line 147: we designed highly flexible thin structures located at the device tip Nothing is said about the resulting compliance of these structures (which would depend on both dimensions and Young's modulus of each material). However, this compliance is known to strongly influence the adhesion force (Gilet et al., *Soft Matter* 15, 3999, 2019). Therefore, it should be quantified and the corresponding elastocapillary number should be compared to that of either fly or beetle seta tips.

Our study proposed the development of a new method to produce an adhesive device that mimics the process of setal formation in *Drosophila*. We agree that the materials used for the generation of

the device and the resulting compliance influence the adhesiveness of the device. We have indicated this in the revised manuscript (line 214-217).

“The compliance of the structure is known to strongly influence the adhesion force. Further validation of our adhesive device, including quantification of the compliance, should be carried to compare the adhesiveness of the device with that of insect footpads.”

However, the quantification of compliance is beyond the scope of this submission. Our lab is currently working on the characterization of the new adhesive device, and the results from this study will be included in a separate report.

6. Line 273: A cross-sectional TEM image of a footpad hair The authors should specify the approximate location of the cross-section: is it across the seta tip or across the stalk?

The location of the cross-section is across the stalk. We have revised the figure legend of **Figure 1** accordingly.

7. Line 414: The ends of the fibres were fixed to the supporter at 2 mm intervals. What was the relative orientation of the fibres before immersion? Was this orientation locally maintained after immersion? Were the fibres clamped at both extremities or only at one?

At first, two nylon fibers at 2 mm intervals were fixed on a paper at one extremity. The fibers were then dipped into a sodium alginate solution and then lifted up from it. After solidification, the spatula was cut out from paper substrate.

We have added this explanation in the Methods section and its schematic diagram in Figure 4.

8. Line 416: a spatulate structure was formed by the surface tension of the solution It would be nice to refer to the existing literature on fibres deformed by surface tension, e. g. Duprat and Protiere, EPL 111, 56006 (2015) or Duprat et al., Nature 482, 510 (2012).

Thanks you for the suggestion. We have included the references in the revised manuscript.

9. Line 421: Adhesive testing of fly-mimetic hair structures was performed Is it normal adhesion or shear adhesion? In figure 3e, it is obviously a shear load at play (i.e., tangent to the contact area). Has the normal adhesion be tested? How do the shear stress and the shear force per unit width compare to results obtained for similar experiments on beetles? (e.g. in Bullock and Federle, J. Exp. Biol. 212, 1876, 2009)

In this study, only the shear test was performed. Normal adhesion means peeling in a perpendicular direction in the case of a tape-like adhesive structure. Since our device is weak against peeling in a perpendicular direction, easy exfoliation by changing the angle is achieved like in an insect.

So, we have added the following sentence in the revised discussion (line 174-176)

“Since our device is weak against peeling in a perpendicular direction, this adhesive ability is easily reversed by applying an opposite directional force or by simply twisting the fiber, similar to what is observed in insect footpads.”

Also, based on the reviewer’s suggestion, we have included a comparison between the result of the shear test in our device and that obtained from similar experiments on beetles. (see, line 169-174)

“Friction and adhesion of single pads were measured in beetles, which suggested that the distal pads containing spatula-tipped hairs, for the most part, are mainly used for pulling and adhesion. So, comparison with the value of the shear stress of distal pads on a smooth surface showed that the shear stress of our device, 784 ± 144 kPa, is comparable to that of distal pads on a smooth surface, 642 ± 99 kPa in males and 613 ± 111 in females.”

10. There are several spelling / grammar mistakes, including in the abstract. Some careful proofreading is needed

We have carefully revised the manuscript to exclude spelling/grammar mistakes and have also asked a language editing service to proofread the manuscript.

Reviewer #3 (Remarks to the Author):

The authors highlight the role of actin in the formation of shape and structure of setae with a spatula shaped end. This shape of foot hairs are of interest from many years due to the interest

in adhesion of insects and geckos to smooth vertical surfaces. This has also prompted a significant interest in developing synthetic structures for robotics and other applications.

The authors have provided some very intriguing results of the formation of setae and with the staining we can learn about the formation of the setae and perhaps the shape of the tip.

I want the authors to address these questions and make changes in the manuscript where applicable.

1. Provide some quantitative analysis of the shape of the spatula and how some statistics on how many spatula were malformed due to Act5C knock-out.

We quantitatively analyzed how many hairs were malformed due to Act5C knock-out. Malformed (forked) tips were seen in $82.2 \pm 5.6\%$ (mean \pm s.e, n=10) and $75.3 \pm 2.3\%$ (n=6) in *svp>Act5C-RNAi*, *GAL80^{TS}* and *svp>Act5C-RNAi*, *elav-GAL80*, respectively. On the other hand, no abnormal tips were observed in wild-type (CS) flies. These results are included in Figure 2f.

2. Elaborate what other functions have been affected by Act5C knock-out other than just the malformed spatula. Based on the movies on how the structure is formed, it is even surprising that even the setae are formed when you do the Act5C knock-out. It appears from the evidence provided that the actin filaments could be also playing an important role in retraction of cells since the tips are remaining in the same place.

We compared the morphology of adult footpad and the formation processes in the pupa between Act5C knock-down flies and wild-type ones. No effects other than just the malformed spatula were seen in the footpad morphology of the Act5C knock-down flies. As the reviewer pointed out, the actin filaments could be also playing an important role in retraction of cells, but our observation showed that the retraction processes and formation of hair stalk in the Act5C knock-down flies are similar to those in the wild-type. Probably this would be due to that the level of knock-down (but not knock-out) in our conditions was insufficient to induce the effects other than the malformed spatula, and the formation of the tips were more sensitive to the knock-down of Act5C.

We have described our observation in the result and discussion section (line 127-130) and added SFigure 5 and SMovie 3 to illustrate this.

“No effects other than the malformed tips were seen in footpad morphology or the process of footpad formation in the Act5C knockdown flies under our experimental conditions (SFigure 5, SMovie 3), probably because the level of knockdown was not high enough to induce other effects.”

3. Have the authors stained for lipids? In the gecko literature there is evidence for lipids covering the setae (Sci. Rep. 2015, 5, 9594).

We did not stain for lipids. However, the footpad secretion is known to contain a non-volatile, lipid-like substance in several insects. So this comment with references was included in the revised Introduction section (line 48-50).

“The footpad secretion is known to contain a non-volatile, lipid-like substance. The bridge between the setal tip and the substrate, filled with the secreted liquid generates capillary forces that allow the insects to adhere to the substrate.”

4. Can the authors comment on the presence of keratin forming proteins.

Drosophila melanogaster does not have a keratin gene. So, there are no interactions of lipid and keratin as in the gecko setae.

5. The authors mention in many places that the tips are softer. There is no mechanical data provided and this has to be corrected.

We changed the word “soft” to “gel”.

6. The analysis of synthetic structures are very preliminary. No discussion is provided on what role the shape plays. What is the role of capillary forces. More discussion needs to be provided.

Based on the reviewer’s suggestion we added the following discussion (line 203-210)

“It is known that the stiffness of insect hair is not uniform, for example, a material composition gradient and subsequent Young’s modulus gradient in the longitudinal direction of the hairs is observed as in the ladybird beetle footpad. The base of hair stalk is sufficiently hard to prevent clustering, and the contact surface of hair tips is sufficiently compliant to adapt to rough substrates. In addition, the bridge between the hair tip and the substrate is filled with the secreted liquid, which generates the capillary forces that adhere the insect to the substrate. In the adhesive device, as in the biological setae, softness at the tip is important for adhesion. ”

Additionally, we also added the following sentence to the Introduction section (line 49-50) and Result and discussion section (line 207-209), respectively.

“The bridge between the setal tip and the substrate, filled with the secreted liquid generates capillary forces that allow the insects to adhere to the substrate.”

“In addition, the bridge between the hair tip and the substrate is filled with the secreted liquid, which generates the capillary forces that adhere the insect to the substrate.”